# A Global Structure and Adaptive Weight Aware ICP Algorithm for Image Registration

Lin Cao [1,2], Shengbin Zhuang [1,2], Shu Tian [1,2], Zongmin Zhao [1,2,*], Chong Fu [3], Yanan Guo [1,2] and Dongfeng Wang [4]

1   The Key Laboratory of Information and Communication Systems, Ministry of Information Industry, Beijing Information Science and Technology University, Beijing 100101, China; charlin@bistu.edu.cn (L.C.); kral154818641@bistu.edu.cn (S.Z.); shutian@bistu.edu.cn (S.T.); yananguo@bistu.edu.cn (Y.G.)
2   The Key Laboratory of the Ministry of Education for Optoelectronic Measurement Technology and Instrument, Beijing Information Science and Technology University, Beijing 100101, China
3   School of Cfiguromputer Science and Engineering, Northeastern University, Shenyang 110169, China; fuchong@mail.neu.edu.cn
4   Beijing TransMicrowave Technology Company, Beijing 100080, China; wdf@tsmtc.com
*   Correspondence: zhaozongmin@bistu.edu.cn

**Abstract:** As an important technology in 3D vision, point-cloud registration has broad development prospects in the fields of space-based remote sensing, photogrammetry, robotics, and so on. Of the available algorithms, the Iterative Closest Point (ICP) algorithm has been used as the classic algorithm for solving point cloud registration. However, with the point cloud data being under the influence of noise, outliers, overlapping values, and other issues, the performance of the ICP algorithm will be affected to varying degrees. This paper proposes a global structure and adaptive weight aware ICP algorithm (GSAW-ICP) for image registration. Specifically, we first proposed a global structure mathematical model based on the reconstruction of local surfaces using both the rotation of normal vectors and the change in curvature, so as to better describe the deformation of the object. The model was optimized for the convergence strategy, so that it had a wider convergence domain and a better convergence effect than either of the original point-to-point or point-to-point constrained models. Secondly, for outliers and overlapping values, the GSAW-ICP algorithm was able to assign appropriate weights, so as to optimize both the noise and outlier interference of the overall system. Our proposed algorithm was extensively tested on noisy, anomalous, and real datasets, and the proposed method was proven to have a better performance than other state-of-the-art algorithms.

**Keywords:** iterative closest point; robust registration; adaptive weight loss metric; global structure; remote sensing image registration

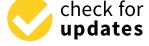



## 1. Introduction

Point cloud registration (PCR) is an important fundamental 3D vision technique, with wide applications in robotics, photogrammetry, and remote sensing contexts, such as simultaneous localization and mapping (SLAM), scene perception, and 3D modeling [1]. The Iterative Closest Point (ICP) algorithm has recently shown great promise in the field of remote sensing image registration [2]. This innovative technique allows for the accurate alignment of multiple images, even when they have significant geometric differences or changes in perspective. Current point cloud acquisition devices mainly focus on how to capture part of an object on a single frame. Therefore, overlapping frames should be taken from different positions to cover the entire object or scene. Point cloud registration is a technique to merge this sequence into a panorama.

ICP [3] is an approach to the standard point cloud registration problem, comprising four main steps, including: initial matching, which selects a point cloud as the reference point cloud, and performs preliminary point-to-point matching between the point cloud to

be matched and the reference point cloud; weighting strategy, which calculates weights based on the distance from each point to the nearest neighbor point, and uses a weighted least squares method for registration; termination condition, which stops the iteration once the error between the current two iterations is less than the preset threshold, and then outputs the registration result; and optional steps, wherein subsequent optimization processing can also be carried out, after the registration step. The above four steps are carried out alternately until the optimal local alignment is achieved. At present, the improvement methods of most ICP algorithms are reflected in the following two aspects: the first is that the convergence area of the ICP algorithm is small, and the convergence speed is slow; the second lies in the sensitivity of the ICP algorithm to outliers, missing data, and partially overlapping regions [4]. This may lead to the incorrect alignment of the ICP algorithm, and an overlapping alignment of some areas in the presence of outliers. The classic ICP algorithm [5] proposes a linear convergence speed, which leads to the slow convergence speed of the algorithm. The point-to-plane mapping relationship improves upon the original ICP algorithm [3], but the function-zero point set of its point-to-plane metric is only a plane patch. The above process exposes the problem of a small convergence domain. Other registration methods have achieved faster convergence. For example, Chen Y et al. [6] proposed a Point-Plane ICP method, which achieved the alignment by minimizing the distance from the point to the plane, while Pottmann H et al. [7] minimized the local quadratic approximation of the squared distance function. Another problem with the ICP algorithm is that, when looking for the best alignment relationship, alignment accuracy can be affected by things such as noise, outliers, and partially overlapping values. The above problems occur frequently in the course of recent research. In response to the above problems, Li J, et al. [8] proposed a new symmetrical point-to-plane distance measurement method, whose function-zero set was a local second-order set surface. It had a wider convergence domain and faster convergence than point-to-point metrics, point-to-plane metrics, and even primitive symmetric metrics. Zhang J, et al. [9] introduced a robust error metric based on Welsch functions, and efficiently minimized it with the MM algorithm with Anderson acceleration. The robust ICP algorithm [10] had certain variant advantages compared to the sparse ICP algorithm [11], to a certain extent. This process was manifested by introducing a robust cost function (instead of loss) to enhance the alignment step.

In this paper, we have proposed a novel and simple method to solve point cloud registration. Compared with the classic ICP algorithm, this study intended to optimize it from the perspective of the following characteristics. Our method was dedicated to optimizing the constraint conditions in order to achieve better convergence performance and faster convergence speed. For partially overlapping area points, we hoped to optimize error discrimination by introducing an adaptive robust loss algorithm. This paper mainly selected representative laser points for matching, and we built a global structure mathematical model based on reconstructing local surfaces. To overcome the problem of a small convergence domain in ICP algorithm, the model took into account the curvature and the normal vectors (i.e., rotation, translation), without increasing the complexity of the algorithm. The metric mathematical model of the local surface was adopted, because the function-zero point set of the point-to-plane metric was a local second-order surface. Algorithms that introduced adaptive robust losses bridged the gap between the nonrobust $\ell 2$ cost and the robust M estimates. This loss was optimized based on a dynamic policy, making it robust to varying degrees, as well as iterative as a function.

The contributions of this paper can be summarized as follow:

(1) This paper introduced a novel ICP variant, GSAW-ICP, incorporating a mathematical model of the global structure to account for the effects of deformation on both the normal vectors and the curvature of the object. The paper has also proposed two innovative metrics: (OAKV) Overlap Area Knockout Value, and (GT) Ground truth interior points, which were used to optimize the convergence strategy.

(2) This paper introduced a loss measurement method based on the adaptive weight adjustment. The method was able to assign appropriate weights to outliers and

overlapping values, as well as optimize the system performance under noise and outlier interference. The method improved the robustness of GSAW-ICP's ability to estimate the gap between the $\ell2$ cost and the robust M.

(3) This paper presented a simulation and testing of the proposed method on the EPFL dataset and a reality measured dataset, before comparing it with the state-of-the-art algorithms. The paper has been organized as follows: Section 2 reviews the related work and the recent improvements of the ICP algorithm; Section 3 describes the solution process of GSAW-ICP and the mathematical model of the global structure; Section 4 explains the convergence criterion and the update iteration of GSAW-ICP, in addition to providing a feasibility analysis of the algorithm; Section 5 reports the experimental results and analysis for GSAW-ICP; and Section 6 concludes the paper. Figure 1 shows the technical flow chart of this paper, where the blue arrows indicate the method flow, and the yellow arrows highlight the novel contributions we made.

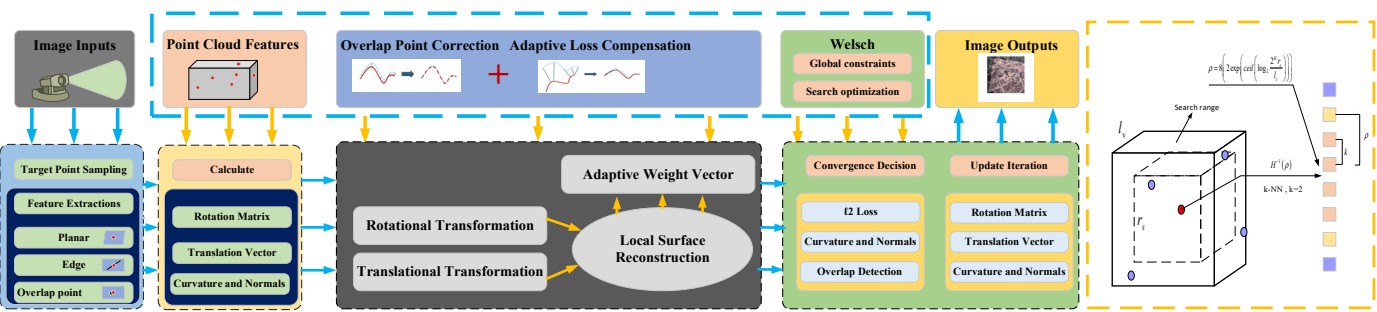

**Figure 1.** GSAW-ICP algorithm flow chart.

## 2. Related Work

The ICP (Iterative Closest Point) algorithm is a widely used technique for aligning two 3D point clouds, and has been studied extensively in the field of computer vision and robotics. Some variations to this algorithm are as follows. Adaptive weighted ICP algorithm based on local features: the traditional ICP algorithm often requires global optimization, resulting in huge amount of calculation. This new method limits the matching to the search radius based on local descriptors, and uses a weighted strategy based on a sparse matrix update to achieve fast, accurate, and robust point cloud registration. Adaptive Weighted ICP Algorithm Combining Deep Learning: by extracting the local features of point clouds using deep learning technology, and combining adaptive weighted ICP algorithms for point cloud registration, this method can cope with complex point cloud shape changes and noise interference, greatly improving the efficiency and accuracy of point cloud registration. The adaptive weighted ICP algorithm based on hierarchical strategy divides the point cloud into multiple levels for registration, and uses high-level information to guide low-level registration, achieving fast and accurate registration. Adaptive weighted ICP algorithm combined with other algorithms: Matching with other algorithms, such as SIFT, SURF, etc., improves the efficiency and accuracy of point cloud registration. However, several challenges still exist in the current state of research. One of the main issues is the sensitivity of ICP to initial alignment, which can lead to suboptimal results or convergence to a local minimum. Various methods have been proposed to overcome this problem, such as using feature-based correspondences or incorporating priors on the transformation. A 3D extension of image matching, feature-based registration likewise involves the two phases of feature matching and geometric estimation [12]. In feature matching, the 3D extension initially realizes via feature detectors [13], such as KeypointNet, MeshDoG, and USIP, before passing through intrinsic shape features (ISS) [14–16]. The important points are given high discrimination by encoding each one into a small feature vector based on the descriptors, and examining its local surface histograms (FPFH) [17], 3DMatch [18], 3DSmoothNet [19], FCGF [20], and SpinNet [21]. Finally, pairwise similarity [22] and chi-square test [23] calculations are used to determine the one-on-one correspondence.

Another challenge is the computational complexity of ICP, especially for large-scale point clouds. Although several optimizations have been proposed, such as hierarchical or parallelized implementations, the scalability of ICP remains a limitation for practical applications. A significant amount of work has reportedly been undertaken to increase the robustness of classical ICP in order to solve this issue. In their pruning technique, Chetverikov et al. [24] suggested preserving selected point pairs with the shortest distances, while discarding other data after each iteration. The rigid registration problem was presented by Granger and Pennec [25] as a maximum-likelihood estimation, and was resolved using the EM algorithm. To lessen the effect of noise and outliers, another probability technique (based on Gaussian mixture representation [26]) has been employed.

Furthermore, ICP assumes that the correspondences between the two point clouds are one-to-one, which may not be true in some scenarios, such as in cases of partial overlap or occlusion. Several extensions of ICP have been proposed to handle such cases, such as nonrigid registration or using probabilistic models, but they are still an active area of research. Several approaches for handling extremely high outliers have recently been put forth. For instance, making sure that outlier removal (GORE), which is reliable for 99% of outliers, finds the genuine upper and lower boundaries of outlier based on the disagreements between the following two. In such cases, we would aim to transform the efficiency index complexity from $o(2^N)$, which exhibits a certain level of complexity, to polynomial complexity $o(N^2)$, through the utilization of two innovative concepts, namely the correspondence matrix and the augmented correspondence matrix. These novel notions enable us to establish stringent boundaries, which can accurately capture the computational characteristics of the problem at hand. A weighted q-norm estimation uses either truncated least-squares (TLS) [27], single point RANSAC [28], or an improved M-like robust estimation to address the scaling, rotation, and translation estimation subproblems of the 7-DoF/6-DoF registration issue [29], in order to condense the parameter space. Despite the good performance, the alignment accuracy of these methods is not as good as the point-based alignment. As a result, feature-based registration is frequently referred to as coarse registration, which leaves point-based registration a good initialization refinement method.

The most popular technique for geometric estimation is the rigid estimate of transformed random sampling results (RANSAC), with six degrees of freedom (DoF) based on robust fitting technology [30] and its derivatives. The model fitting and random sampling steps are alternated during the threshold convergence. However, the computing complexity of the the RANSAC-type approaches grows exponentially with the rate of exceptions. The 3D feature matching algorithm is substantially more challenging than the Scale Invariant Feature Transform (SIFT) and Radiation-variation Insensitive Feature Transform (RIFT) algorithms; because 3D feature matching algorithms can better solve problems, such as uneven density, lack of texture and noise. As a result, there are many outliers in the initial response set (often >95%). In this situation, it could take tens of minutes (or possibly hours) for RANSAC type approaches to arrive at a rough solution, which makes them impractical.

To address these challenges, recent studies have proposed novel approaches to ICP, such as using learning-based methods, or integrating semantic information. For example, deep-learning-based approaches have been used to learn robust features or to predict correspondences between the two point clouds. Another approach is to incorporate semantic information, such as object-level or category-level knowledge, to guide the alignment process.

Overall, while ICP has been a fundamental technique in 3D point cloud processing, there are still many challenges and opportunities for further research and innovation. Based on the discussion of the above problems, this paper has summarized the shortcomings of the above methods. When dealing with point cloud registration of remote sensing images, there will be more disturbances, such as outliers and overlapping values in the space. These problems will lead to a reduction in registration accuracy, which will further affect the detection rate of the overall spatial information. For objects whose shape needs to be determined in advance, if ground objects have a fixed shape (such as airplanes, ships, houses, etc.) then the accuracy of remote sensing images can still be guaranteed. In the

case of significant deviations, such as in woodland, the characteristics of dense and sparse textures vary. Furthermore, extensive research has been conducted on employing the ICP algorithm in the context of multisensor data fusion. This application involves the alignment of data acquired from distinct sensors, such as optical and radar sensors. This can potentially improve the accuracy of remote sensing applications, such as land cover classification and change detection. Based on the findings of the above problems, when designing the ICP point cloud registration method for remote sensing applications, this paper fully considers the judgment of the method's convergence area caused by the object's deformation. At the same time, the designed method should have a good adaptive weight iteration process to adapt to the processing of outliers and overlapping values in complex spatial remote sensing scenarios.

## 3. Classical ICP Revisition

The problem of calibrating two point clouds is to find the rotation and translation that maximize the overlap between the two clouds. In this chapter, the classic ICP [3] algorithm will be introduced.

### 3.1. Iterative Nearest Point

The fundamental tenet of the ICP algorithm is that the original point cloud $P$ and the target point cloud $Q$ must match within specific bounds. Finding the nearest neighbor point $(p_i, q_i)$ is the first step in determining the best matching parameters $R$ and $t$ in order to produce the best error function. The following limitations apply:

$$E(R, t) = \frac{1}{n} \sum_{i=1}^{n} \| q_i - (R p_i + t) \|^2 \tag{1}$$

where $p_i$ is the point from the source point cloud $P$, $q_i$ is the point corresponding to $Q$ in the target point cloud $p_i$ to be matched, $R$ is the rotation matrix, and $t$ is the translation vector. ICP has developed a wide range of versions that vary in the information matrix $\omega_{ij}$ or the heuristics used to detect correspondences, enhancing their resilience and performance over time. This issue can be simplified to the original ICP formulation with a condensed Euclidean metric between related points, using $\omega_{ij}$ as the identity. One must choose an $\omega_{ij}$ such that the point-to-plane metric is minimized, and that its eigenvalues along the normal direction are all zero. The acquisition of the original point set, identification of the related point set, computation of the transformation matrix, and comparison of whether the distance between the two points is larger than the threshold are the important components of the algorithm:

$$\| p_i^c - T \oplus p_j^r \| > \varepsilon_d \tag{2}$$

As illustrated in Figure 2, the two sets of solution point clouds should totally overlap. Space remote sensing involves a lot of noise interference, overlapping scan point values, and missing values. These issues may result in corresponding mismatching, as well as having a significant impact on the success or failure of alignment. As a result of the corresponding noise and mismatching, the two correspondences are not the same point in the corresponding space, though the algorithm will treat them as such. As a result, Equation (1) does not always hold, leading to the introduction of the corresponding minimization objective function, which is followed by the solution of the associated rotation matrix and translation vector problem.

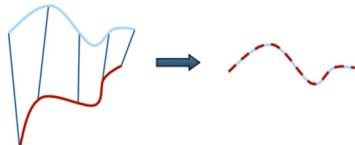

**Figure 2.** Ideal case of point cloud alignment.

In two sets of point sets, $P = \{x_1, \cdots, x_a\}$ and $Q = \{y_1, \cdots, y_b\}$ provided by $\mathbb{R}^d$, under specific restrictions. To determine the ideal matching parameters $R$ and $t$ so as to produce the ideal error function, as well as locate the nearest neighbor point $(p_i, q_i)$, the following constraint equation can be used:

$$\min_{R,t} \sum_{i=1}^{a} \left( \min_{y \in Q} \| R x_i + t - y \| \right)^2 + I_{so(d)}(R) \tag{3}$$

where $I_{so(d)}(\cdot)$ is the indicator function formula of the special orthogonal group $so(d)$. The definition of $I_{so(d)}(\cdot)$ is Equation (4), and $\min_{y \in Q} \| R x_i + t - y \|$ is the distance from the transformation point $R x_i + t$ to the target set $Q$. The translation vector $t$ and rotation matrix $R$ are both listed as parameters for this function.

$$I_{so(d)}(R) = \begin{cases} 0, if\, R^T R = I\, \text{and}\, \det(R) = 1 \\ +\infty, otherwise \end{cases} \tag{4}$$

where $I$ is a matrix of $\mathbb{R}^{3 \times 3}$ and $\det(\cdot)$ is a determinant of a matrix.

*3.2. Iterative Nearest Point*

When choosing point sets for feature distance determination, the ICP algorithm and some ICP improved algorithms use representative point sets rather than point sets that are randomly chosen. This can reduce the amount of calculation to some extent, while also reducing the outlier (uneven distribution) of point sets that is brought on by the offset. However, the above problem suffers from a small convergence domain, as well as from a poor robustness of the algorithm. The total registration effect and model effect will be impacted by this. We discovered through investigation that the point cloud concealed the true surface. These structured points might have been chosen as representative points. The aforementioned example locations possessed improved normal vectors and curvature, contributing to a more effective constraint on the model. Smaller dots were observed to exhibit higher curvature. Moreover, the structured points fulfilled the smaller convergence domain constraints imposed by the model. The selecting process should accommodate the points' balance and observability. If the aforementioned conditions are not taken into account, it is crucial to consider the observability of the point cloud image in both direction $X$ and direction $Y$. Neglecting this aspect can lead to a change in the resulting outcome. We built a mathematical model of the global structure with the following definitions:

$$I^{P_k}(x) = \frac{\sum_{p_i \in P_k} W_i(x)((x - p_i) \cdot \vec{n}_i)}{\sum_{p_j \in P_k} W_j(x)} \tag{5}$$

Weight A is defined as:

$$W_i(x) = e^{-\|x - p_i\|^2 / h^2} \tag{6}$$

The hidden surface of point cloud set $P_k$ was represented by $I^{p_k}(x) = 0$, while its subimage was made up of data from the previous $n$ frame, and its point $p_i$ normal vector was represented by $n_i$. The hidden surface of the point cloud set $P_k$ was located at a distance of $I^{P_k}(x)$ from point $x$ in space $\mathbb{R}^3$. The purpose of weight $W_i(x)$ was to only choose a portion close to point $x$, such that Formula (5) could be used to reconstruct the surface. The weight $W_i(x)$ gradually decreased as the distance from point $x$ on the surface increased. The projection $y_i$ of matching solution point $x_i$ on the surface was:

$$\overline{x}_i = R x_i + t, \overline{y}_i = \overline{x}_i - I^{P_k}(\overline{x}_i) \vec{n}_i \tag{7}$$

For a new frame of point cloud data $S_k$, solve the transformation matrix by minimizing the following function:

$$\sum_{x_i \in S_k} \left( (Rx_i + t - y_i) \cdot \vec{n}_i \right)^2 \tag{8}$$

A fresh frame of data was represented by $S_{k+1}$, $I^{P_k}(x_i)$ was the distance from the surface to point $x_i$ in the current frame $S_k$, and $\vec{n}_i$ was the normal vector of the point closest to point $x_i$ in frame $P_k$. The benefit of GSAW-ICP was that it could produce robust results without the necessity for feature extraction and point cloud segmentation. When a solid beginning value was assured, it had strong accuracy and convergence. GSAW-ICP fully utilized the point cloud structure information, eliminating the original ICP technique that is prone to running into the local optimal problem, as opposed to merely reflecting on the point-to-point and point-to-plane distance limitations in the constraints.

### 3.3. Point Cloud Alignment Process

The point cloud centralization phase, which involves determining the locations of the centers of two sets of point clouds, must be carried out using the original ICP algorithm. All of the point pairs included in the calculation are from the set $C$, meaning that $C$ is the total number of related points.

$$\mu_Q = \frac{1}{|C|} \sum_{(i,j)} q_i, \mu_P = \frac{1}{|C|} \sum_{(i,j) \in C} p_j \tag{9}$$

Aligning the coordinate system's origin requires subtracting each point's matching point cloud center and unifying the coordinate system.

$$Q' = \{q_i - \mu_Q\} = \{q'_i\}, P' = \{p_j - \mu_P\} = \{p'_j\} \tag{10}$$

The goal function is then solved using an orthogonal Procrustes method Formula (11):

$$\begin{aligned} E(R,t) &= \sum_{(i,j) \in C} \|q_i - Rp_j - t\|^2 \\ \|X\|_F &= \sqrt{trace(X^T X)} = \sqrt{\sum_{i,j} x_{ij}^2} \end{aligned} \tag{11}$$

where the first item to the right of the equal sign indicates that the point cloud is extracted from $Q$ and organized in matrix form, the points in the $P$ matrix are transformed by the rotation matrix $R$, and both the rotation matrix $R$ and the error function $E(R, t)$ only take into account the rotation matrix $R$. equal to the difference between is:

$$E'(R) = \left\| [q'_1 \cdots q'_n] - R[p'_1 \cdots p'_n] \right\|_F^2 \tag{12}$$

The 2-norm calculates the square of the value, which is then solved using SVD. The $F$-norm determines the difference between corresponding points. Make the covariance matrix by computing:

$$W = \sum_{(i,j) \in C} q'_i p'^T_j \tag{13}$$

Singular values, which are regarded as the representative values of the matrix, can be utilized to represent the information contained in the matrix when using SVD to compress image data. The singular value represents more information the greater it is. Therefore, you simply need to restore the top few greatest single numbers in order to restore the data itself. Find the target point graphically as shown in Figure 3.

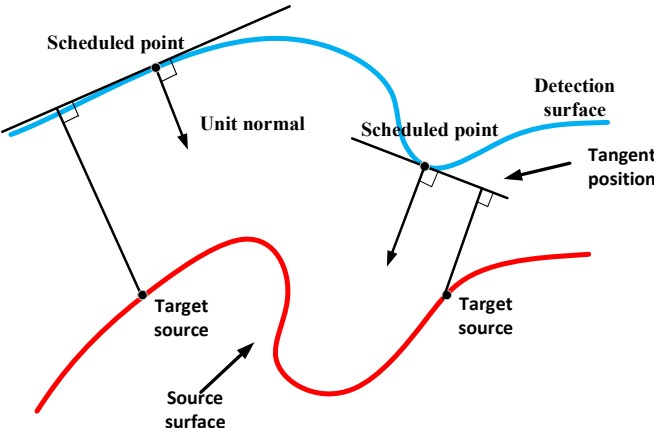

**Figure 3.** Find the target point graphically.

We've used a graphic to illustrate this relationship. It is typically impossible to identify the appropriate relative rotation and translation in one step without knowing the correct correspondence. As a result, it is important to assume that two unknown pieces of data are related and to iterate constantly. In Figure 4, the alignment diagram is displayed.

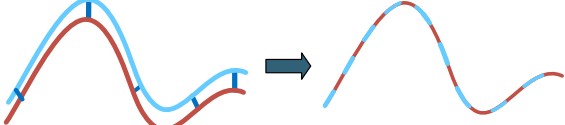

**Figure 4.** Iterative alignment schematic.

The error is taken to be limitless throughout the process of finding a solution. Determine the point that corresponds to the error that exceeds the specified threshold, compute $R$ and $t$ using SVD, apply $R$ and $t$ to the point that has to be aligned, compute the error function, and iterate continuously. End the iteration when there is no longer any change in the relationship. By adjusting the construction error function's difference, we were able to decrease the number of sampling points, and by sampling the normal vector space, we can guarantee the upward continuity of the normal vector, better preserve regions with evident curvature changes, and smooth out regions with sparse features.

The number of sampling points was high where the curvature varies in Figure 5, which could help to ensure that the object's properties were not lost. Here, the benefits of GSAW-ICP were further illustrated. For areas with significant changes in curvature, normal vector space sampling was more aligned. By examining whether this portion of the area formed a full figure and the thickness of the associated item, we were able to determine the impact of the algorithm on the circular area, in accordance with Figure 6.

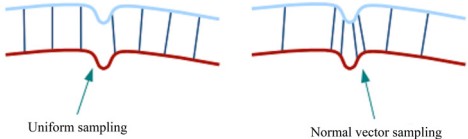

**Figure 5.** Schematic diagram of uniform sampling and normal vector sampling.

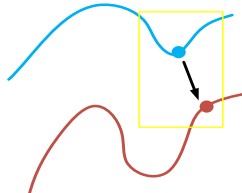

**Figure 6.** Schematic diagram of initial value iteration.

The point set to be matched was intersected with the GSAW-ICP, which was projected along the point normal vector. The schematic diagram of normal vector projection is shown in Figure 7. Smooth structural entities produced slightly better convergence results than other ICP techniques. Due to curvature constraints weights, the convergence outcomes for noise or complicated structures (overlapping points and outliers) will be determined, in order to achieve the optimization impact.

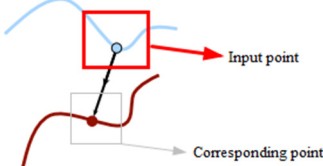

**Figure 7.** Schematic diagram of normal vector projection.

GSAW-ICP could recognize anomalous spots and adaptively set the threshold. The threshold was set to blue correlation in Figure 8 because the blue value was less than the red value. The selection was rejected when the feature restrictions between nearby point pairs were considerably different, as seen in Figure 9.

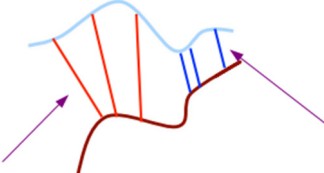

**Figure 8.** Schematic diagram of exception handling.

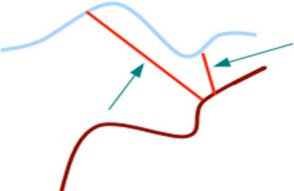

**Figure 9.** Schematic diagram of large differences in feature constraints.

### 4. Loss Metrics for Adaptive Weight Adjustment

We have primarily introduced a loss metric method that made use of adaptive weight modification in this section. In the experimental dataset, the method efficiently and adaptively modified the corresponding weights for anomalous, overlapping, and noisy data. The first subsection explains the justification and problem statement for developing adaptive weight measurements. We explain how the rotation matrix $R$, the translation vector $t$, and the iterative update for adaptive weight adjustment were derived in the second subsection. The viability of the suggested strategy is examined in Section 3 of this chapter. Designing an adaptive weight metric that is effective and robust to the various forms of data in the experimental dataset is the main objective of this section.

#### 4.1. Dealing with Outliers

Given that the $\ell_2$ distance is used as a criterion, classical ICP may produce results with outliers and partially overlapping mismatches. In this part, we suggest the GSAW-ICP algorithm, an adaptive weight-based robust ICP algorithm. GSAW-ICP can automatically perform sparse learning to fix the weights of related point samples by using adaptive neighborhood weight learning, increasing the robustness of the ICP algorithm.

GSAW-ICP has two phases, similar to conventional ICP. Finding the correlation between the source point set and the target point set, where $c_z(i)$ is the starting point and $z$ is

the current iteration number, is the first stage. The first step's problem is the same as the basic ICP problem, which can be stated as follows:

$$
\begin{aligned}
c_z(i) = \ &\operatorname{argmin} \| (R_{z-1} p_i + t_{z-1}) - q_{c(i)} \|^2 + I^{P_k}(x) \\
&\text{s.t.} c_z(i) \in 1, 2, \ldots, N_q, i = 1, \cdots, N_p
\end{aligned} \tag{14}
$$

The K-D tree algorithm approach or an effective closest point search method can be used to solve the aforementioned formula. Robust rigid body transformation calculations are the major focus of GSAW-ICP. The corresponding formula is represented as follows:

$$
\begin{aligned}
\mathcal{L}(R, t, w) = \ &\operatorname{argmin} \sum_{i=1}^{n} \left( w_i \| (R p_i + t) - q_{c(i)} \|^2 + \gamma \| w_i \|^2 \right) \\
&\text{s.t.} \sum_{i=1}^{n} w_i = 1, 0 \leqslant w_i \leqslant 1, R^\top R = I_d, \det(R) = 1
\end{aligned} \tag{15}
$$

### 4.2. Update Iterative Process

In this section, we suggest an iterative method to maximize the objective function Formulation (15) presented in the formulation above. For the rotation matrix $R$, the translation vector $t$, and the adaptive weight vector $w$, we have provided the derivation of the update iteration process for the above parameters.

- Update the rotation matrix:

When fixing $t$ and $w$, remember that Formula (14) only has one term. Given that Formula (14) and $R$ are connected, we can write it as follows:

$$
\begin{aligned}
\| R x_i - y_i \|^2 \ &= (R x_i - y_i)^\top (R x_i - y_i) = \left( x_i^\top R^\top - y_i^\top \right)(R x_i - y_i) \\
&= x_i^\top R^\top R x_i - y_i^\top R x_i - x_i^\top R^\top y_i + y_i^\top y_i \\
&= x_i^\top x_i - y_i^\top R x_i - x_i^\top R^\top y_i + y_i^\top y_i.
\end{aligned} \tag{16}
$$

where $x_i^\top R^\top y_i$ is a scalar. Therefore, the following results can be obtained:

$$
x_i^\top R^\top y_i = y_i^\top R x_i, \tag{17}
$$

Rewrite Equation (17) as follows:

$$
\| R x_i - y_i \|^2 \ = x_i^\top x_i - 2 y_i^\top R x_i + y_i^\top y_i \tag{18}
$$

We removed the unnecessary expression of $R$ by substituting Formula (18) for Formula (14), and this approach had no impact on the minimization solution when the formula was pushed.

$$
\begin{aligned}
R \ &= \operatorname{argmin} \sum_{i=1}^{n} -2 w_i y_i^\top R x_i + \gamma \| w_i \|^2 \\
&= \operatorname{argmin} \sum_{i=1}^{n} w_i y_i^\top R x_i.
\end{aligned} \tag{19}
$$

According to the properties of the rotation matrix, the form of the Formula (20) is:

$$
\sum_{i=1}^{n} w_i y_i^\top R x_i = tr\left( W Y^\top R X \right) = tr\left( R X W Y^\top \right) \tag{20}
$$

Among them, $W = diag(w_1, w_2, \ldots, w_n)$. $S = X W Y^\top$ is the weighted covariance matrix in this instance. Singular value decomposition should then be applied to $S$, to yield:

$$
S = U \sum V^\top \tag{21}
$$

Substituting Equation (21) into Equation (20) gives:

$$tr\left(RXWY^\top\right) = tr(RS) = tr\left(\Sigma V^\top RU\right) \tag{22}$$

The matrix $M = V^\top RU$ is orthogonal because, according to Formula (22), $R$, $U$, and $V$ are all orthogonal matrices. $M$ confirms that the matrix's rows and columns are all orthogonal vectors, then $|m_{ij}| \leqslant 1$. Representing the diagonal elements of the singular value matrix as $\sigma_1, \sigma_2, \ldots, \sigma_d \geqslant 0$, we can derive:

$$tr\left(\Sigma V^\top RU\right) = tr(\Sigma M) = \sum_{i=1}^{d} \sigma_i m_{ii} \leqslant \sum_{i=1}^{d} \sigma_i \tag{23}$$

The best solution of Formula (19) can be obtained when $m_{ii} = 1$, as demonstrated using Formula (23).

$$M = I = V^\top RU \Rightarrow R = VU^\top \tag{24}$$

$R$ is an orthogonal matrix that can be utilized as a rotation matrix or a reflection matrix in Formula (24). By applying the following conditional Formula (25), we can tell them apart.

$$\begin{cases} \det\left(VU^\top\right) = -1, R \text{ is the reflection matrix} \\ \det\left(VU^\top\right) = +1, R \text{ is the rotation matrix} \end{cases} \tag{25}$$

$R$ might be a rotation or reflection matrix if the two point clouds are coplanar but not collinear; these possibilities will be covered and examined later. The following form $D = diag\left(1, 1, \det\left(VU^\top\right)\right)$ can be used to express it for processing, and the rotation matrix can be calculated as follows:

$$R = VDU^\top, \tag{26}$$

- Update the translation vector:

When $R$ and $w$ are fixed, the partial derivative of $t$ in Formula (15) is:

$$\begin{aligned}\frac{\partial \mathcal{L}}{\partial t} &= \sum_{i=1}^{n} 2w_i\left(Rp_i + t - q_{c(i)}\right) \\ &= 2t\left(\sum_{i=1}^{n} w_i\right) + 2R\left(\sum_{i=1}^{n} w_i p_i\right) - 2\sum_{i=1}^{n} w_i q_{c(i)}\end{aligned} \tag{27}$$

Taking the partial derivative of Formula (27) as 0, we can obtain:

$$t = \bar{q} - R\bar{p}, \bar{p} = \frac{\sum_{i=1}^{n} w_i p_i}{\sum_{i=1}^{n} w_i}, \bar{q} = \frac{\sum_{i=1}^{n} w_i q_i}{\sum_{j=1}^{n} w_i} \tag{28}$$

- Update the adaptive weight vector:

The $i$th point cloud pair's registration error is expressed as follows:

$$e_i = \| (Rp_i + t) - q_{c(i)} \|^2 \tag{29}$$

Solving $w$ is comparable to optimizing the following issue when $R$ and $t$ are fixed:

$$\min_{0 \leqslant w_i \leq 1, \mathbf{w}^\top \mathbf{1} = 1} \sum_{i=1}^{N} w_i e_i + \gamma \| w_i \|^2 \tag{30}$$

After algebraic operation, we can obtain:

$$\min_{0 \leqslant w_i \leq 1, \mathbf{w}^\top \mathbf{1} = 1} \frac{1}{2} \| w + \frac{e}{2\gamma} \|_2^2 \tag{31}$$

The constraint problem's Lagrangian function is thus defined as follows (31):

$$L(w, \lambda, \sigma) = \frac{1}{2} \parallel w + \frac{\mathbf{e}}{2\gamma} \parallel_2^2 - \lambda \left( w^\top - 1 \right) - \sigma^\top w \tag{32}$$

where $\lambda \in \mathbb{R}$ and $\sigma \in \mathbb{R}^\top$ are the respective scalar and vector components of the Lagrange coefficient. Since the constraints include inequality constraints, KKT (Karush–Kuhn–Tucker Conditions) conditions must be met in order to solve the constraints using the Lagrange multiplier method. The following KKT requirements are met by Formula (32):

$$\begin{cases} \frac{\partial \mathrm{L}(w, \lambda \sigma)}{\partial(w)} = 0 \Rightarrow w_i + \frac{e_i}{2\gamma} - \lambda - \sigma_i = 0 \\ w_i \geqslant 0 \\ \sigma_i \geq 0 \\ w_i \sigma_i = 0 \end{cases} \tag{33}$$

Differentiating $w_i$ and setting it to zero, we obtain:

$$w_i = \lambda + \sigma_i - \frac{e_i}{2\gamma} \tag{34}$$

It is noteworthy that we can reach the best solution of $w_i$ under the KKT condition $w_i \sigma_i = 0$:

$$w_i = \max \left( \lambda - \frac{e_i}{2\gamma}, 0 \right) \tag{35}$$

$w$ should be bound by $k$ nonzero components because our registration approach tends to learn a sparse weight vector $w$ to totally remove the detrimental influence of excessively noisy data. Without loss of generality, we assume alignment errors $e_1 \leqslant e_2, \ldots, \leqslant e_n$, in ascending order. We have $w_k > 0$ and $w_{k+1} = 0$, since $w$ only has $k$ nonzero components. The expression that results is as follows:

$$\begin{cases} w_k > 0 & \Rightarrow \lambda - \frac{e_k}{2\gamma} > 0 \\ w_{k+1} \leq 0 & \Rightarrow \lambda - \frac{e_{k+1}}{2\gamma} \leq 0 \end{cases} \tag{36}$$

The expression of parameter $k$ can be determined if parameter $\lambda$ is set as its upper bound, in accordance with Formula (36), and the corresponding constraint conditions are represented as $\mathbf{w}^\top = 1$:

$$\sum_{i=1}^{N} \left( \lambda - \frac{e_i}{2\gamma} \right) = 1 \Rightarrow \lambda = \frac{1}{N} \left( 1 + \sum_{i=1}^{N} \frac{e_i}{2\gamma} \right) \tag{37}$$

Then, the result of integrating Formulas (36) and (37) can be deduced:

$$\frac{k}{2} e_k - \frac{1}{2} \sum_{i=1}^{k} e_i < \gamma \leqslant \frac{k}{2} e_{k+1} - \frac{1}{2} \sum_{i=1}^{k} e_i \tag{38}$$

The parameter $\gamma$ is set as its upper bound in order to obtain the best solution with exactly $k$ nonzero terms $w_i$:

$$\gamma = \frac{k}{2} e_{k+1} - \frac{1}{2} \sum_{i=1}^{k} e_i \tag{39}$$

We can directly solve $w_i$ using the following strategy, substituting the parameters $\lambda$ and $\gamma$ from Formula (37) into Formula (40):

$$w_i = \begin{cases} \frac{e_{k+1}-e_i}{ke_{k+1}-\sum\limits_{j=1}^{k} e_j}, i \le k \\ 0, i > k \end{cases} \tag{40}$$

This technique successfully distinguishes outliers from noise, as illustrated in Formula (40); when $i \le k$, the weight corresponding to the bigger alignment error is allocated to a smaller value. It has been demonstrated that the GSAW-ICP method can effectively manage noise and outliers from a theoretical perspective. When $i > k$, we can reduce extreme noise point pairs by setting the value of $w_i$ to zero, such that only $k$ point pairs with the smallest alignment fault are taken into account for registration. With regard to space remote sensing photos, the aforementioned method's design may effectively address the issues of outlier, noise, and outlier processing.

### 4.3. Update Iterative Process

The accuracy and robustness of registration are to be improved, in addition to improving the processing of noise, outliers, and overlapping values in spatial remote sensing images, using the novel ICP algorithm (GSAW-ICP) that has been proposed in this study. We performed a feasibility analysis for the GSAW-ICP algorithm's primary improvements, with the following results:

To describe a curvature- and normal-vector-based model, the rigid body transformation model (which has typically been used in traditional ICP methods) was predicated on the idea that the relative positional relationship between the target point cloud and the reference point cloud never changes. Although the item may be bent, stretched, or other forms of deformation in actual applications, these deformations frequently influenced the accuracy and robustness of registration. The curvature- and normal-vector-based model we have presented takes object deformation into account, which could more accurately characterize the deformation relationship between the target point cloud and the reference point cloud, increasing registration accuracy. Related Evidence Section 4.2 provides theoretical justifications.

On the use of an adaptive weight adjustment in a loss metric model, by excluding comparable points with significant mistakes or positional biases, some ICP variations achieved resilience. However, numerous studies have demonstrated that these techniques are challenging to fine-tune for successful outcomes. These techniques might also have made it more likely to become caught in local minima. In contrast, our approach employed a dynamic threshold strategy to gradually penalize outliers. Our registration model performed better in both accuracy and robustness, compared to robust ICP. This was made possible by our adaptive, robust model, which was essential to the registration procedure. Our method's update iteration process had been demonstrated on a theoretical level, which gave the algorithm solid theoretical backing, as shown in Section 4.2.

The most popular techniques for geometric estimation are the six-degrees-of-freedom rigid-based robust fitting methodology estimates transformed random sampling results (RANSAC) and their derivatives. Model fitting and random sampling stages alternate during threshold convergence. However, the computing complexity of RANSAC-type algorithms increases exponentially with the outlier rate. The 3D feature matching algorithm is substantially more challenging to use than the SIFT and RIFT algorithms; the causes of this are uneven density, a lack of texture, and noise. As a result, there are many outliers in the initial response set (usually >95%). RANSAC-type algorithms can take tens of minutes (or even hours) to arrive to a rough solution in this situation, which makes them impractical.

With the improvements in the aforementioned two areas, our algorithm operated more reliably and effectively while processing spatial remote sensing data. For the fourth chapter, we conducted a substantial number of tests and comparative analyses to confirm the efficacy

and superiority of our algorithm, in order to confirm its viability. To further demonstrate the feasibility and academic nature of the algorithm, we also combined theoretical analysis and model generation.

## 5. Experimental Results and Analysis

Herein, we have first evaluated the surface reconstruction model of GSAW-ICP's convergence domain and convergence rate in comparison to standard point-to-point and point-to-surface distance metrics. Next, we assessed how GSAW-ICP manages outliers and partial overlaps. We contrasted the GSAW-ICP method with a few more sophisticated algorithms in terms of index comparison; RSME, K value, goodness of fit (R2), a total sum of squares of deviation (SST), and temporal comparison were the primary topics covered. Additionally, we creatively suggested two assessment indicators to assess the algorithm's overall impact: the alignment value of estimated inliers and ground truth inliers, and the value of eliminating overlapped areas. Two new metrics have been helpful in evaluating how well the algorithm performs. We finished up by running a simulation experiment. We started by rendering a simple point cloud registration on the data set. The impacts of anomalous processing were then compared for various data that contained noise and outliers, in order to show the GSAW-ICP's robustness. Finally, we showed the registration effect in various scenarios using real point cloud data.

The more traditional EPFL dataset [31], the KITTI dataset [32], and self-test data were used in our experiment. The operating system used the Intel(R) Core(TM) i7-6200U CPU, 2.4 GHz, and 8GB RAM, together with MATLAB R2020a, Python3.8, C++, Eigen, MATLAB point cloud data box, PCD, and other tools. The Swiss Federal Institute of Technology produced the open EPFL dataset with the goal of offering high-quality, massive, multidomain data resources for scholarly research. This data set included data sets from computer vision, natural language processing, machine learning, biomedicine, physics, and other domains, in addition to including image, text, video, and other media, as well as biological and physical data. The broad application domain and good data quality of the EPFL dataset were its distinguishing features. To guarantee the accuracy, completeness, and reproducibility of the data, the data collecting and processing methodology for this dataset closely adhered to the standards and procedures of scientific research. The dataset also supported numerous data formats and multilingual data, giving researchers a more practical and adaptable approach to using the data. There is also a video dataset under this dataset, including over 4000 hours of video data. A popular computer vision dataset called KITTI included a lot of stereo images, laser point clouds, camera calibration parameters, and other information. This dataset, which was produced collaboratively by Germany's Max Planck Institute and the Karlsruhe Institute of Technology, was primarily utilized for autonomous vehicle research. This dataset included a wide range of settings, including urban, rural, highway, etc., with various types of roads, weather, and traffic. The KITTI dataset was a collection of around 45,000 RGB and grayscale photographs. The shooting angles from front to back were 0 degrees, +40 degrees, and −40 degrees, and the size of each image was $1242 \times 375$ pixels. Lidar point cloud data from various road scenarios (including point cloud coordinates, reflectivity, and labels) were included in the KITTI collection. Tens of millions of data points with three coordinate values and a reflectance value made up the total amount of data. Additionally, the KITTI dataset comprised the internal and external camera calibration parameters, such as the camera's focal length, the location of the optical center, the rotation matrix, and the translation vector. Data from images and point clouds could be transformed into 3D space using these parameters.

Perform point cloud registration experiments on monkey models from the EPFL statue dataset, we first compared the registration results. In more detail, the source point cloud was made up of the first 60% of the whole model's points, while the target point cloud was made up of the remaining 60%. A rigid transformation ($[0°, 20°], [20°, 40°], [40°, 60°]$) was applied on the target set at random, with the overlap of this pair only being about 33. The experimental results are presented in Figure 10 to demonstrate the advantages of our

suggested model based on curvature and normal vectors over point-to-point and point-to-plane models, as well as the rotation invariance of the GSAW-ICP algorithm. Among the eight approaches that were examined, our GSAW-ICP obtained the best RMSE accuracy while having a significantly shorter running time. Despite the ICP method's excellent robustness, the small convergence pool prevented it from achieving good registration. No initialization data was supplied throughout this registration process. Compared to ICP, ICP-l, and its accelerated variations, symmetric ICP performed significantly better. To eliminate outliers, it relied on distance and normalcy restrictions. It could therefore withstand outliers and partial overlaps.

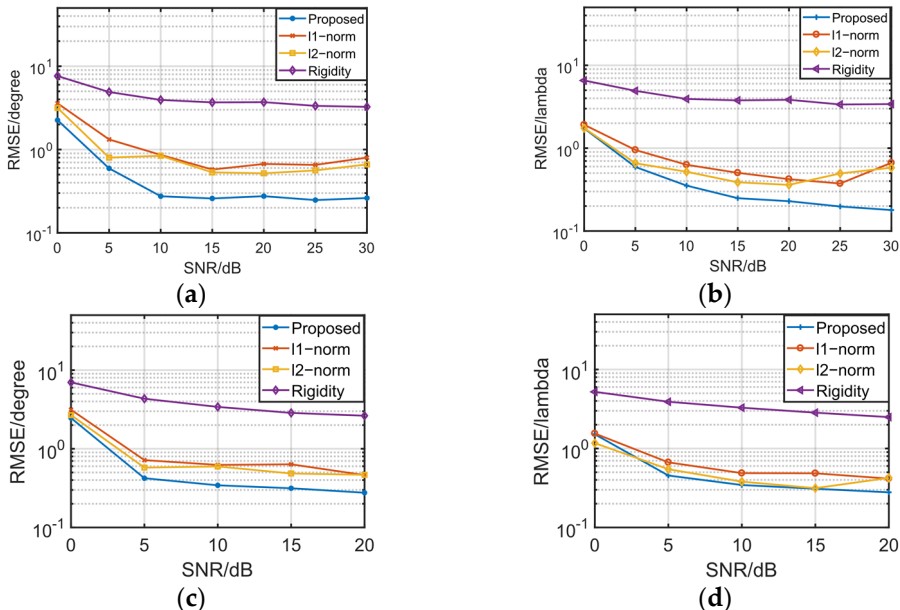

**Figure 10.** The relationship between the root mean squared error of the registration effect and the signal-to-noise ratio. From (**a**–**d**), the number of iterations increases by 10, while (**a**) has 50 iterations.

Table 1 shows the outcomes of our registration trials using four spatial remote sensing point cloud data that were registered under the KITTI dataset. The name of the comparison algorithm is listed in the first column, and the number is listed in the second. To determine the error between the registration effect and the genuine value for each piece of data, we calculated the central target point of the corresponding point cloud data. The distance error in columns 4 and 5 and the error rate in the center were used to calculate the error; the lower the value, the better. It is evident from the trials in Table 1 that the GSAW -ICP algorithm outperformed the other four approaches in terms of distance error and center error rate, as well as its ability to manage the registration of spatial remote sensing data.

The comparative results of the elimination values in the overlapped regions of various algorithms are shown in Table 2. This result indicates that the elimination value of the initial model could not satisfy the ideal model iteration of the iterative convergence technique. The more information that could be employed in the original model, the higher the elimination value was. The feature description was worse the less points it had. To some extent, it served as a reflection of the quality of the constraint convergence model. The alignment values of the interior points were estimated using various techniques, and the actual interior points are shown in Table 3. There were discrepancies between the actual mapping and registration procedure and the true value of the space object after the image mapping relationship of the sensor. It was possible to determine the mapping transfer error of the registration process by comparing the alignment of the estimated inliers output from the registration results with the object's ground truth, which served as an excellent example for enhancing the model's mapping, calibration, and registration.

**Table 1.** Comparison results of initial registration center and real data center.

| Algorithm | Num | Dataset 1 | | | Dataset 2 | | | Dataset 3 | | | Dataset 4 | | |
|---|---|---|---|---|---|---|---|---|---|---|---|---|---|
| | | Center Coordinates | Dis (m) | DER (%) | Center Coordinates | Dis (m) | DER (%) | Center Coordinates | Dis (m) | DER (%) | Center Coordinates | Dis (m) | DER (%) |
| ICP | 1 | (2.136, 37.054) | 0.644 | 5.04 | (2.768, 42.473) | 3.317 | 15.89 | (6.309, 50.831) | 0.621 | 3.69 | (−1.811, 39.733) | 1.082 | 7.89 |
| | 2 | (3.002, 35.207) | 1.38 | 10.81 | (2.613, 36.703) | 2.002 | 9.59 | (4.193, 45.645) | 1.373 | 8.17 | (−0.732, 35.737) | 3.029 | 22.34 |
| | 3 | (6.199, 41.047) | 0.421 | 3.30 | (7.632, 47.041) | 0.558 | 2.67 | (5.706, 39.449) | 0.337 | 2.01 | (2.740, 38.105) | 0.927 | 6.84 |
| AA-ICP | 1 | (1.636, 35.959) | 1.013 | 7.93 | (2.459, 41.404) | 2.483 | 11.89 | (6.304, 49.974) | 0.336 | 2.00 | (−2.178, 39.789) | 0.824 | 6.07 |
| | 2 | (3.990, 37.417) | 1.116 | 8.74 | (3.031, 35.781) | 2.740 | 13.13 | (3.565, 44.726) | 0.313 | 1.86 | (−0.487, 35.942) | 2.718 | 20.5 |
| | 3 | (6.331, 41.545) | 0.135 | 1.06 | (7.264, 46.621) | 0.788 | 3.77 | (5.891, 38.915) | 0.902 | 5.36 | (2.664, 38.137) | 0.992 | 7.32 |
| Sparse ICP | 1 | (1.697, 35.626) | 1.352 | 10.59 | (0.166, 40.232) | 0.144 | 0.69 | (6.550, 49.360) | 0.883 | 5.25 | (−2.910, 39.365) | 0.230 | 1.69 |
| | 2 | (3.170, 37.434) | 1.30 | 10.18 | (3.875, 38.937) | 0.566 | 2.71 | (3.221, 45.164) | 0.534 | 3.18 | (−0.423, 36.913) | 1.962 | 14.47 |
| | 3 | (6.250, 41.242) | 0.227 | 1.78 | (7.603, 46.798) | 0.725 | 3.47 | (5.817, 39.178) | 0.630 | 3.75 | (2.735, 38.065) | 0.894 | 6.60 |
| Robust ICP | 1 | (1.487, 36.552) | 0.411 | 3.22 | (0.412, 40.083) | 0.358 | 1.71 | (6.617, 48.378) | 1.867 | 11.11 | (−3.056, 39.482) | 0.416 | 3.07 |
| | 2 | (3.839, 36.024) | 0.289 | 2.26 | (3.631, 37.058) | 1.415 | 6.78 | (3.301, 46.799) | 2.167 | 12.89 | (−0.273, 38.682) | 1.476 | 10.89 |
| | 3 | (6.222, 41.278) | 0.189 | 1.48 | (6.441, 45.726) | 1.850 | 8.86 | (5.438, 39.942) | 0.226 | 1.34 | (3.226, 38.306) | 1.022 | 7.54 |
| GSAW-ICP | 1 | (1.387, 36.852) | 0.157 | 1.23 | (0.312, 40.483) | 0.156 | 0.75 | (6.221, 50.131) | 0.306 | 1.82 | (−2.856, 39.012) | 0.213 | 1.57 |
| | 2 | (3.839, 36.324) | 0.016 | 0.13 | (3.665, 38.581) | 0.155 | 0.75 | (3.157, 44.775) | 0.179 | 1.07 | (0.073, 38.182) | 1.03 | 7.60 |
| | 3 | (6.322, 41.578) | 0.150 | 1.17 | (7.321, 47.519) | 0.157 | 0.75 | (5.677, 39.523) | 0.258 | 1.53 | (3.226, 38.506) | 1.221 | 9.01 |

**Table 2.** Comparison of running results between GSAW-ICP and several typical registration algorithms.

| Algorithm | Evaluation Index (Overlap Area Knockout Value) | | | | |
|---|---|---|---|---|---|
| | Bimba | Children | Dragon | Angle | Bunny |
| ICP | 25.3 | 32.4 | 27.4 | 28.3 | 33.6 |
| ICP-*l* | 23.2 | 26.1 | 25.1 | 26.8 | 30.2 |
| AA-ICP | 27.7 | 24.6 | 26.3 | 27.1 | 39.4 |
| Sparse ICP | 21.6 | 17.2 | 20.1 | 22.9 | 24.3 |
| Fast ICP | 13.2 | 14.6 | 12.8 | 17.6 | 18.4 |
| Robust ICP | 16.9 | 16.4 | 13.5 | 15.8 | 19.9 |
| Symmetric ICP | 15.7 | 13.1 | 15.3 | 18.2 | 21.5 |
| GSAW-ICP | 11.1 | 10.7 | 12.8 | 15.3 | 18.9 |

Our experiments in Tables 2 and 3 found that the two new constraint variables of (OAKV) Overlap Area Knockout Value and (GT) Ground truth interior points exhibited a better optimization effect on the convergence of the model. During the iteration of the algorithm, the size of the convergence domain determined the number of elimination values of the algorithm, and more elimination values are less effective for the description of the model. The GSAW-ICP algorithm combined the curvature and normal vector of the model to better filter the elimination values and achieve the purpose of improving the alignment. In addition, the Ground truth interior points were able to play a better role in constraining the accuracy of the eigenvalues. In the adaptive weight update phase, Ground truth interior points played a good role in optimizing the deviation values of the model.

**Table 3.** Comparison of running results between GSAW-ICP and several typical registration algorithms.

| Algorithm | Evaluation Index (Ground Truth Interior Points) | | | | |
|---|---|---|---|---|---|
| | Bimba | Children | Dragon | Angle | Bunny |
| ICP | 14.62 | 20.31 | 18.28 | 17.24 | 21.27 |
| ICP-*l* | 15.71 | 21.56 | 17.89 | 13.51 | 22.14 |
| AA-ICP | 18.92 | 23.64 | 23.51 | 20.01 | 26.41 |
| Sparse ICP | 16.21 | 21.28 | 20.37 | 18.29 | 22.41 |
| Fast ICP | 19.27 | 26.83 | 28.41 | 21.25 | 30.58 |
| Robust ICP | 19.89 | 25.89 | 28.22 | 22.18 | 28.89 |
| Symmetric ICP | 18.26 | 22.17 | 26.19 | 21.16 | 28.75 |
| GSAW-ICP | 19.58 | 27.12 | 27.81 | 22.71 | 29.61 |

After the comparative analysis in Tables 2 and 3, we concluded that the GSAW-ICP algorithm had better robustness and accuracy compared to other advanced algorithms. Specifically reflected in the OAKV metric, the performance of the GSAW-ICP algorithm on different models was reduced by about 18% on average, compared to other advanced algorithms. In terms of GT metrics, the GSAW-ICP algorithm improved by about 6% on average. To be able to better illustrate the overall superiority of the GSAW-ICP algorithm, we summarized a ranking system by combining the quantitative analysis in Tables 2 and 3. Under each model, we counted the average ranking of each algorithm. Eight algorithms were rated from one to eight according to their values, with five models under each table. We gave each model a weight of 0.2 in order to determine the ranking of the eight algorithms, which have been arranged as indicated in Table 4. Smaller average ranking values meant better performance.

**Table 4.** Comparison of running results between GSAW-ICP and several typical registration algorithms.

| Algorithm | Average Ranking | | |
|---|---|---|---|
| | OAKV | GT | Mean |
| ICP | 7.6 | 7.6 | 7.6 |
| ICP-*l* | 6.2 | 7.2 | 6.7 |
| AA-ICP | 7.2 | 5.2 | 6.2 |
| Sparse ICP | 5 | 6.2 | 5.6 |
| Fast ICP | 2 | 2 | 2 |
| Robust ICP | 3.2 | 2.2 | 2.7 |
| Symmetric ICP | 3.4 | 4.4 | 3.9 |
| GSAW-ICP | 1.2 | 1.8 | 1.5 |

The data in the above table illustrates that the GSAW-ICP algorithm ranked first overall, compared to other algorithms, with an average ranking of 1.5 in performance metrics under each model. This shows that our algorithm had better accuracy and robustness compared to other algorithms. In second place was the Fast ICP algorithm, which ranked second on average for both performance metrics, and second overall. The outcomes of the GSAW-ICP algorithm ablation experiment have been presented in Table 5. The technical index of the traditional ICP algorithm is in the first row of the table, and the variant index is in the second row. The performance index of the fundamental GSAW-ICP method that we suggested is shown in the third row of data. The loss measuring method of adding adaptive weight adjustment to GSAW-ICP is represented by the data value in the fourth line. The method of adding the constraint model based on surface reconstruction after adding the loss measurement method to GSAW-ICP is represented by the data value in the fifth row. It is clear from comparing the results that the changes we suggested were able to improve the ICP algorithm to different degrees.

**Table 5.** Comparison of running results between GSAW-ICP and several typical registration algorithms.

| Algorithm | Evaluation Index | | | |
| --- | --- | --- | --- | --- |
| | **0.02ADD** | **0.05ADD** | **0.1ADD** | **Mean** |
| ICP | 26.23 | 66.38 | 89.21 | 60.61 |
| ICP-*l* | 32.51 | 72.41 | 91.52 | 65.48 |
| GSAW -ICP | 33.27 | 71.62 | 92.86 | 65.92 |
| +ARC-Welsh | 35.81 | 75.74 | 93.71 | 68.42 |
| +Surface | 42.61 | 78.47 | 95.72 | 72.27 |

In the second portion of the experiment, while other variables stayed the same, we compared the effects of various SNRs on the registration accuracy of various constraints (point-to-point constraint, point-to-plane constraint, and direct transformation). The root mean-squared error of the mean result estimate of 500 separate trials was used in this paper to assess the findings. The range of the signal-to-noise ratio was 0 dB to 30 dB. Figure 10 depicts the link between the signal-to-noise ratio and the root mean square error of the registration effect.

As can be seen in Figure 10, the performance of the algorithm in this paper was better than the other three methods. For the root mean squared error of the alignment results, the performance of the other three methods was similar and tended to be stable. For the RMSE of distance, the RMSEs of the above three methods no longer changed with increasing signal-to-noise ratio because they did not take into account the constraining effect of object shape on the overall convergence. Therefore, the GSAW-ICP method had a higher alignment accuracy compared with the other three methods.

In the third section of the experiment, while other variables remained constant, we compared the effects of various iterations on the registration accuracy of various constraints (point-to-point constraints, point-to-plane constraints, and direct transformation). The root mean squared error of the mean result estimate of 500 separate trials was used in this paper to assess the findings. The range of the signal-to-noise ratio was 0 dB to 30 dB. Figure 11 depicts the association between the number of iterations and the root mean square error of the registration effect.

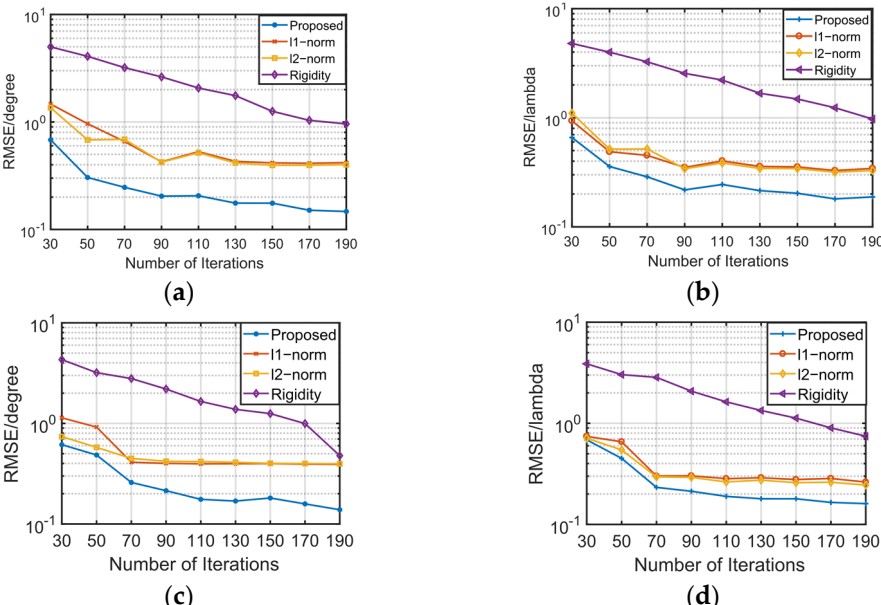

**Figure 11.** The relationship between the root mean squared error of the registration effect and the signal-to-noise ratio. From (**a**–**d**), the noise interference increases by 5 db in sequence, while (**a**) has a noise interference of 0 db.

As can be seen in Figure 11, the performance of the algorithm in this paper was better than the other three methods. For the root mean square error of the alignment results, the performance of the other three methods was similar and tended to be stable. The RMSE of the aforementioned approach stopped changing as the number of repetitions increased until it reached more than 90. The GSAW-ICP method offered a greater registration accuracy when compared to the other three methods. The algorithm's ability to adjust to the speed and effectiveness of processing spatial remote sensing images was improved with fewer iterations.

The fourth part of the experiment was the experiment of the simulation environment. Both 2D and 3D images were used for registration simulation testing; Figure 12 depicts the original image. Figure 13 displays the outcomes of the experiment with the 2D image. As a test set for studies, the original image was rotated and translated at a specific angle. The original image was gray, and the rotated and translated image was light red. Last but not least, we randomly chose four spots from the original image to test the registration impact. On the images registered at various angles and directions, there were matching points corresponding to those points. Figure 13 displays the drawing in question.

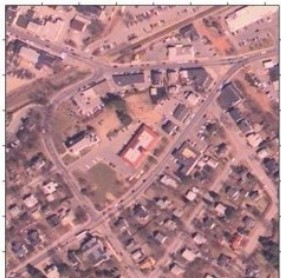

**Figure 12.** Image to be registered.

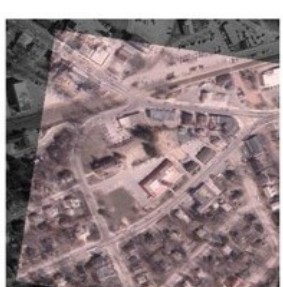 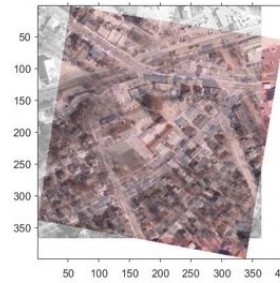 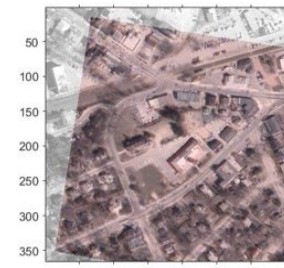 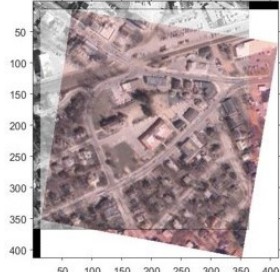

**Figure 13.** Image registration under different rotation angles.

We also applied the technique and rotated the image to find the registration impact of the remote sensing image based on Figure 12. For aerial photography, precise region-specific registration was necessary. Our registration algorithm successfully registered the many reference locations we chose in Figure 12, and as demonstrated in Figure 14, they may be precisely positioned in the actual area.

In addition, we used a 3D scanner to collect real data for point cloud image registration. In this part of the experiment, in order to verify the registration effect of the algorithm in real scenes, three sets of real point cloud images were collected for registration. We conducted registration experiments on the site point cloud images of the three point clouds and the TruSlicer images of the point clouds, respectively. For the convenience of distinction, we added the corresponding blue points as centralized representations, which have been recorded as the original point cloud 1, 2, and 3, respectively. The results are shown in Figures 15–17. The TruSlicer images of the three point clouds are shown in Figures 18–20.

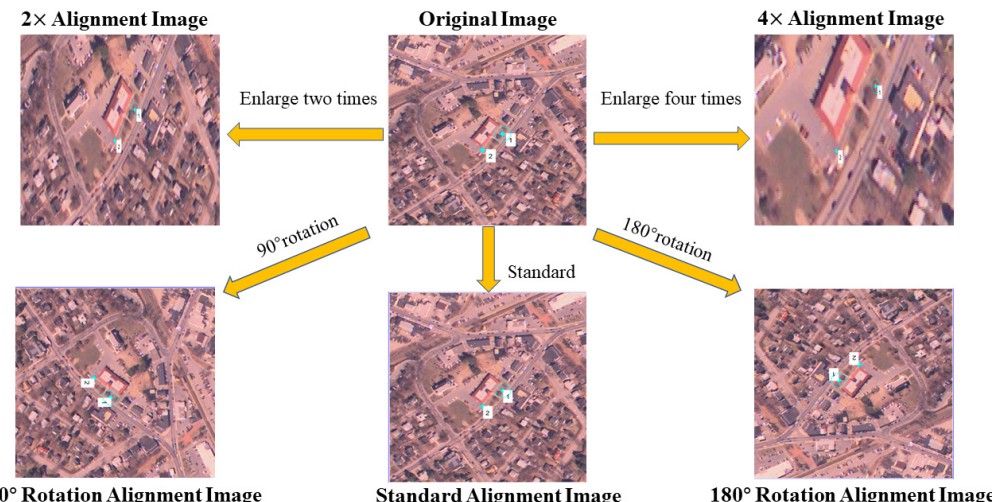

**Figure 14.** Remote sensing registration effect detection.

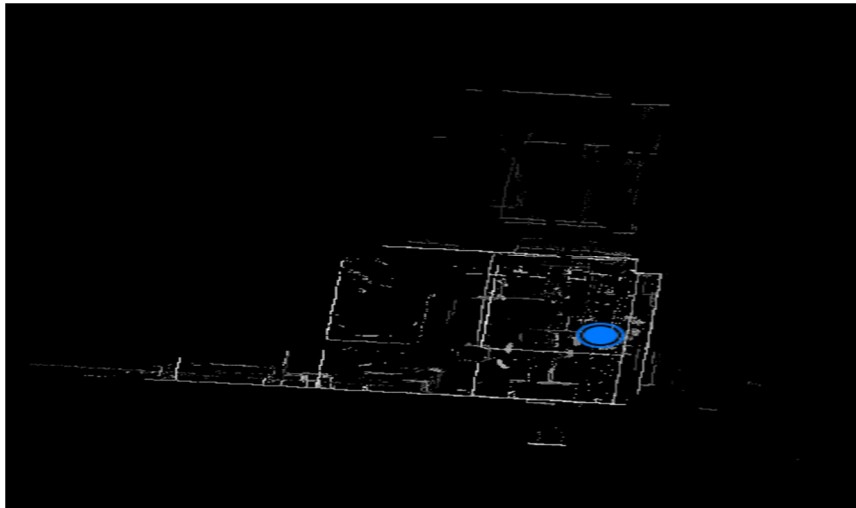

**Figure 15.** Point cloud map of the original site 1.

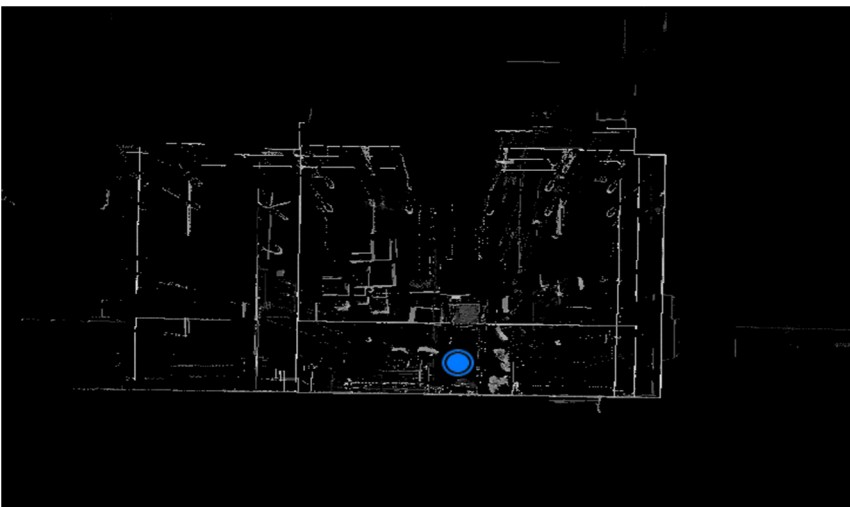

**Figure 16.** Point cloud map of the original site 2.

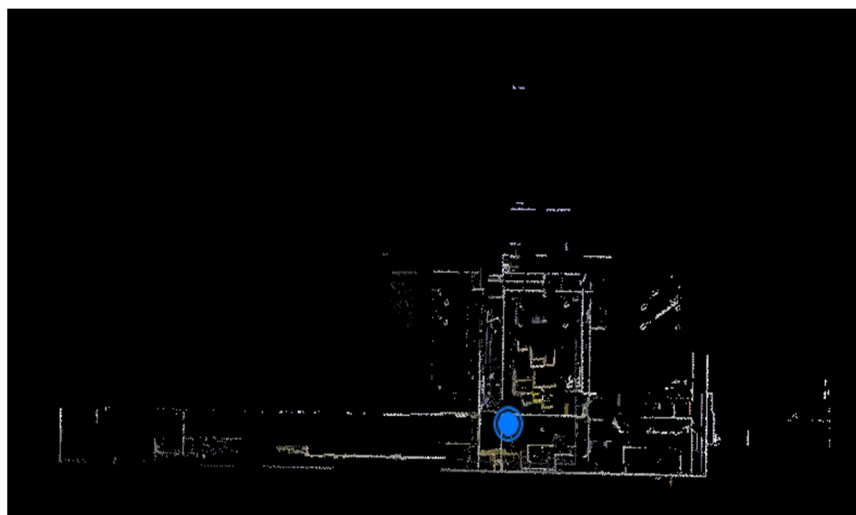

**Figure 17.** Point cloud map of the original site 3.

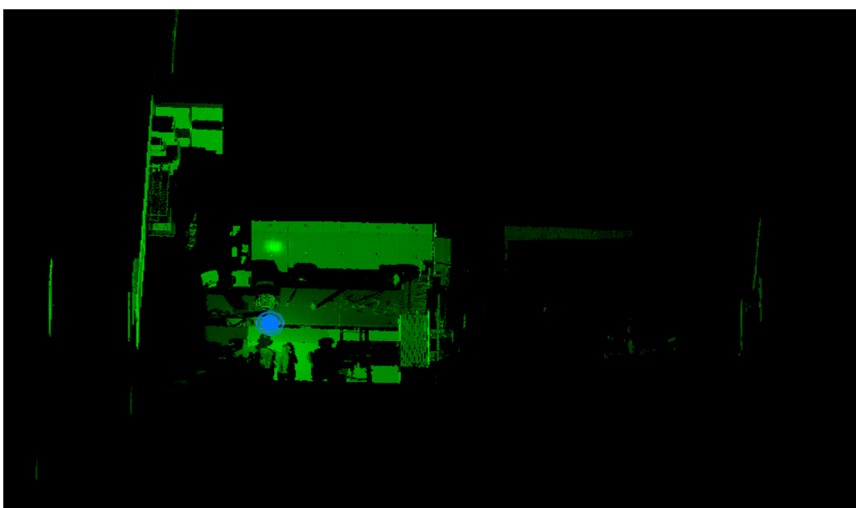

**Figure 18.** Raw point cloud 1TruSlicer image.

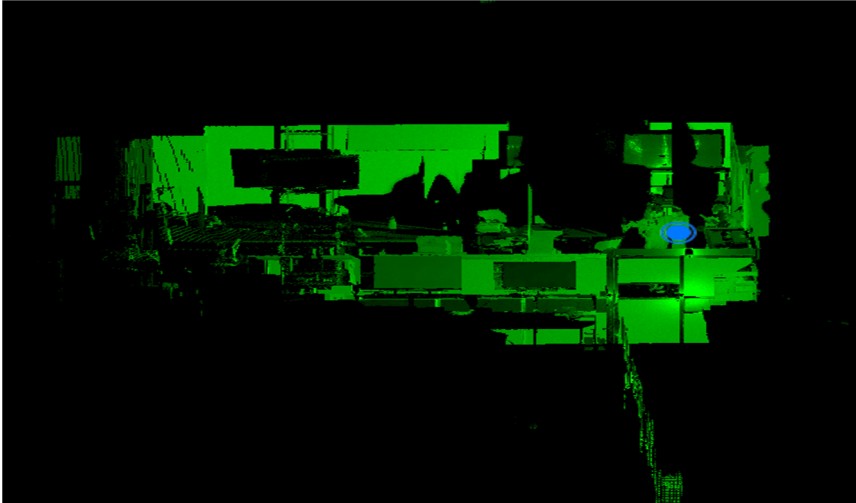

**Figure 19.** Raw point cloud 2TruSlicer image.

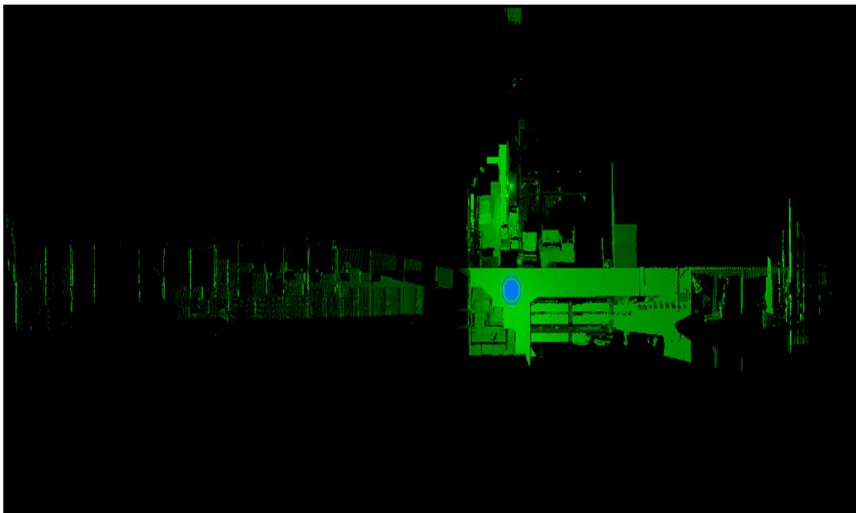

**Figure 20.** Raw point cloud 3TruSlicer image.

The GSAW-ICP algorithm was used for the registration of the above point cloud (labeled 1.2). The effect diagram of the point cloud registration of the site is shown in Figure 21. The red dots in the figure were conveniently selected for observation and recording. After the registration process was completed for the convenience of recording, the figure showed the matching state of the original 1.2 blue spots. The initial 1.2 point cloud TruSlicer picture registration is shown in Figure 22. Figure 23 shows the registration effect of the original site point cloud 2 and site point cloud 3. Figure 24 shows the registration results of the original TruSlicer point cloud 2 and TruSlicer point cloud 3.

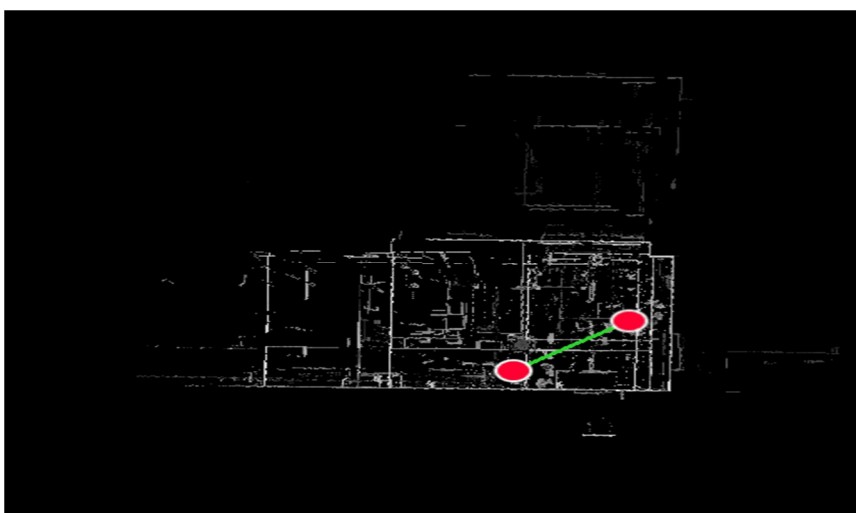

**Figure 21.** Original 1.2 site point cloud registration map.

The above experimental plots were obtained from measurements in a real environment. We have experimentally presented the point cloud plots 1, 2, and 3 under one graph. In Figure 25, we can observe that the red dots used as reference points are registered together. In Figure 26, the TruSlicer image clearly shows the alignment results of the three point cloud maps. The red part of the figure corresponds to the part of the point cloud map 3. The blue part of the figure corresponds to the part of the point cloud map 1. The green part of the figure corresponds to the part of the point cloud map 2.

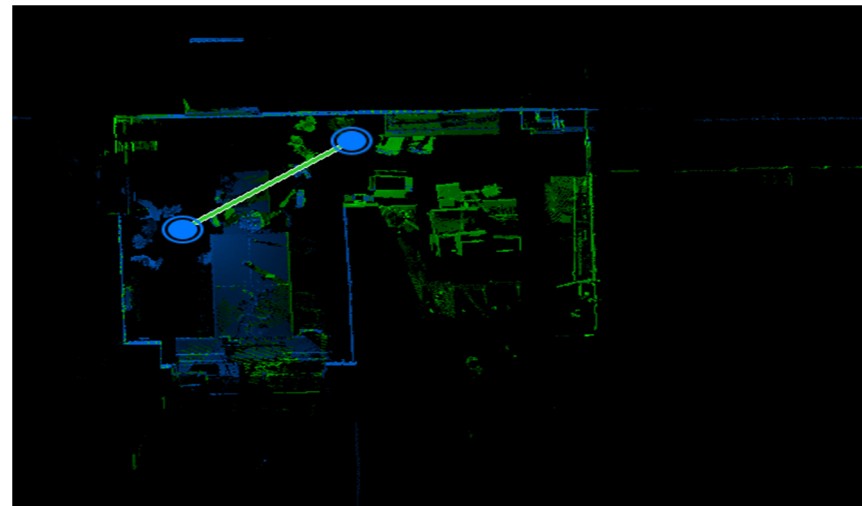

**Figure 22.** Original 1.2 point cloud TruSlicer image registration.

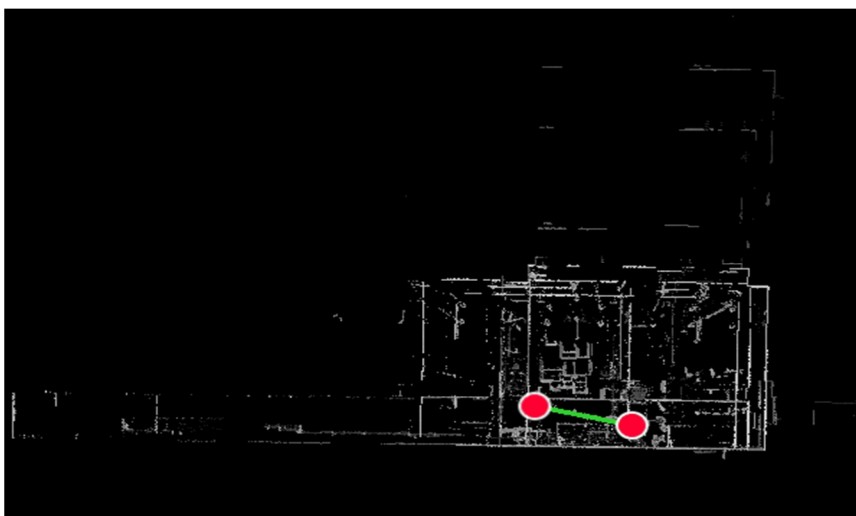

**Figure 23.** Registration of original 2 and 3 site point cloud images.

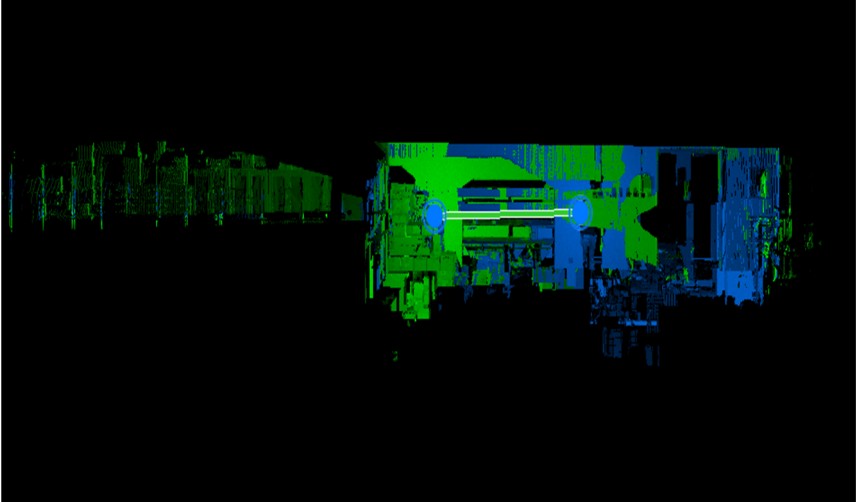

**Figure 24.** Registration of original 2 and 3 point cloud TruSlicer images.

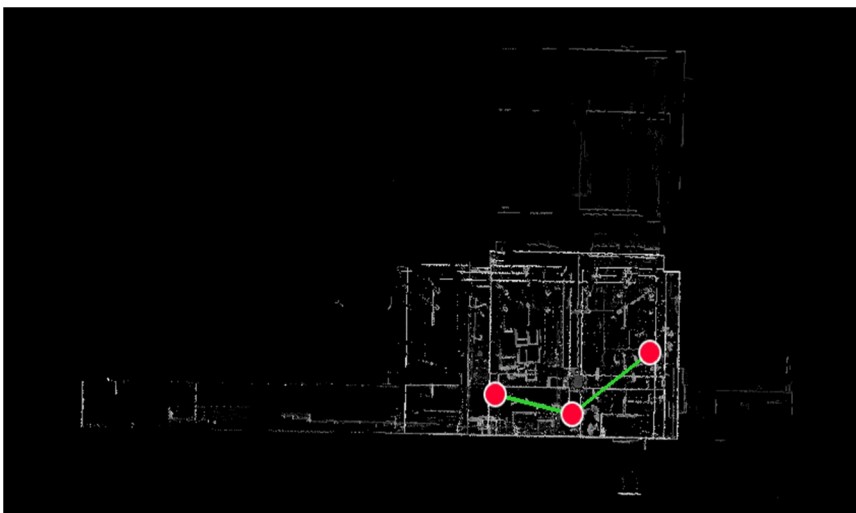

**Figure 25.** Registration of original 1, 2, 3 site point cloud images.

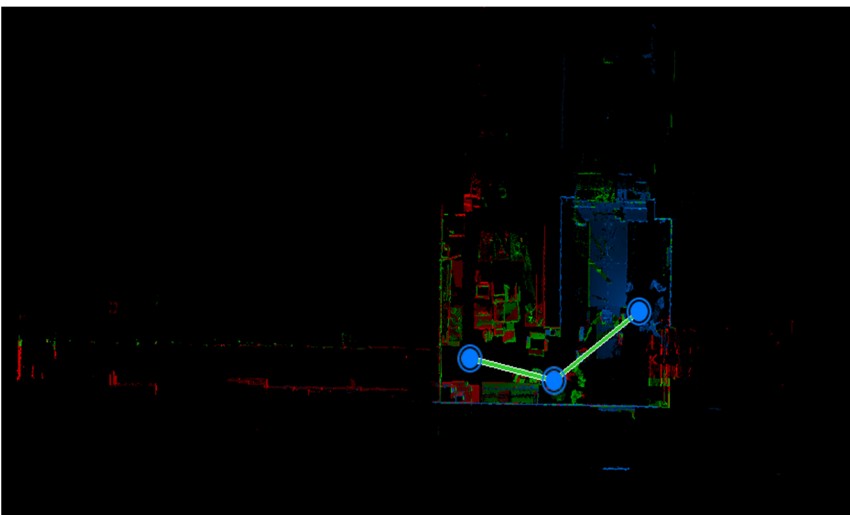

**Figure 26.** Original 1, 2, 3 point cloud TruSlicer image registration.

The aforementioned tests demonstrated that our GSAW-ICP approach could successfully register both 2D and 3D images. In 2D alignment experiments, we verified the rotation invariance and translation invariance of the algorithm by translating and rotating them. The GSAW-ICP algorithm had good alignment, even after two and four times magnification. After different degrees of rotation and translation, the GSAW-ICP algorithm was able to correctly align the control points in the image. We compared the registration errors and deviations of five algorithms, including the classical ICP algorithm. The registration accuracies of these algorithms are presented in Table 1. Our proposed GSAW-ICP algorithm exhibited smaller errors compared to the other algorithms, performing well across all four datasets. This indicates that our algorithm achieved superior registration results. As for the comparison of convergence performance, we can refer to Figures 10 and 11. These figures demonstrate the convergence performance of our proposed algorithm in comparison to other constraint-based convergence methods. The images display the RMSE and SNR of different algorithms as a function of iteration count. From the results, we observed that our proposed method (represented by the blue curve in the figures) demonstrated favorable constraint-based convergence. Furthermore, to validate the rotational and translational invariance of the algorithm, we enlarged the original image in Figure 12 by a factor of two and four, and performed registration after rotating it by 90° and 180°. The results, as shown in Figure 14, confirmed that our algorithm exhibited good rotational and transla-

tional invariance. The validation of the above process helps the algorithm to be applied in real remote sensing scenarios, and our algorithm was still able to provide more stable results when the video sensor was exposed to airflow bumps or other impacts. Under 3D alignment experiments, our algorithm was able to maintain high accuracy alignment in the face of complex point cloud image structures. Figures 15–26 show the swept results of the point cloud images in a real environment with a more complex spatial structure and many disturbances. The GSAW-ICP algorithm was capable of achieving alignment results not only in the planar sense, but also in the face of point cloud sets in space. Many complex conditions may arise in the real use of space remote sensing, and in some cases sensor fusion technology is required to increase the algorithm's data processing level. We believe that the GSAW-ICP approach can handle many issues in the remote sensing environment because it has demonstrated strong robustness and accuracy in the aforementioned studies for a variety of diverse environments and conditions.

## 6. Conclusions

This paper proposed a precise and reliable registration technique based on a mathematical model of the global structure. The model accounted for the rotations and translations of normal vectors and curvatures, resulting in a wider convergence domain and effect. GSAW-ICP outperformed similar advanced methods in terms of registration quality, convergence speed, and robustness to noise, outliers, and partial overlap. The paper also introduced a loss metric with adjustable weights that could be adapted to the point set characteristics and reduce the impact of overlap and outliers. Furthermore, the paper provided a strategy for the iterative update process and a theoretical support for the optimality of GSAW-ICP. The paper conducted extensive experiments on simulated datasets and demonstrated the superiority of GSAW-ICP in registration accuracy and outlier handling. The paper suggested that this method could be applied to high-precision registration tasks (such as ground–air remote sensing).

**Author Contributions:** Conceptualization, L.C., S.Z. and S.T.; methodology, S.Z.; software, Z.Z.; validation, L.C., D.W. and C.F.; formal analysis, Y.G.; investigation, C.F.; resources, L.C.; data curation, Z.Z. and S.Z.; writing—original draft preparation, L.C. and S.Z.; writing—review and editing, S.Z. and Z.Z.; visualization, S.Z.; supervision, C.F.; project administration, D.W.; funding acquisition, Y.G. All authors have read and agreed to the published version of the manuscript.

**Funding:** This work was supported by the National Science Foundation of China under Grant U20A20163, Scientific Research Project of Beijing Municipal Education Commission under Grant KZ202111232049, National Natural Science Foundation of China 62001032.

**Data Availability Statement:** The original datasets are publicly available from The KITTI Vision Benchmark Suite (cvlibs.net) and Statues—Geometric Computing Laboratory (epfl.ch).

**Conflicts of Interest:** The authors declare no conflict of interest.

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
