# Peer review of "A Global Structure and Adaptive Weight Aware ICP Algorithm for Image Registration"

_remotesensing, doi:10.3390/rs15123185_

Round 1

Reviewer 1 Report

The ICP algorithm is an efficient and accurate point cloud registration algorithm with broad application prospects. It can be applied in fields such as robot vision, 3D reconstruction, medical image processing, etc., providing strong support for the development of related technologies.

After carefully reading the manuscript, I would like to put forward some suggestions from my own perspective and communicate with the author. I hope our communication can help improve the quality of the manuscript.

1. In Line 45 of 1. The basic steps of the adaptive weighted ICP algorithm should be four instead of two, including, Initial matching: Select a point cloud as the reference point cloud, and perform preliminary point-to-point matching between the point cloud to be matched and the reference point cloud. Weighting strategy: Calculate weights based on the distance from each point to the nearest neighbor point, and use weighted least squares method for registration. Termination condition: When the error between the current two iterations is less than the preset threshold, stop the iteration and output the registration result. Optional steps: After registration, subsequent optimization processing can also be carried out.

2. In Line 88 of 1. This section should not introduce the research results achieved in the paper, but should be included in the conclusion section.

3. In Line 116 of 2. These contents should be cited. Including,Adaptive weighted ICP algorithm based on local features: traditional ICP algorithm often requires global optimization, resulting in huge amount of calculation. This new method limits the matching to the search radius based on local descriptors, and uses a weighted strategy based on sparse matrix update to achieve fast, accurate and robust point cloud registration. Adaptive Weighted ICP Algorithm Combining Deep Learning: Extracting Local Features of Point Clouds Using Deep Learning Technology, and Combining Adaptive Weighted ICP Algorithm for Point Cloud Registration. This method can cope with complex point cloud shape changes and noise interference, greatly improving the efficiency and accuracy of point cloud registration. The adaptive weighted ICP algorithm based on hierarchical strategy divides the point cloud into multiple levels for registration, and uses high-level information to guide low-level registration, achieving fast and accurate registration. Adaptive weighted ICP algorithm combined with other algorithms: Matching with other algorithms such as SIFT, SURF, etc. improves the efficiency and accuracy of point cloud registration.

4. In Line 202 of 3.1. Please refer to the writing standards for scientific papers.

5. In Line 408 of 4.2. What is the specific value when i=k?

6. In Line 472 of 5. If possible, please show the thumbnails of these datasets.

7. In Line 629 of 5. The text in the bottom right corner of the image is not readable.

8. In Line 698 of 5. Please provide additional explanation on the effectiveness of the algorithm proposed in the paper in improving registration accuracy, accelerating convergence speed, and supporting data registration with different resolutions.

9. In Line 704 of 6. The serial number of the chapter is incorrect.

10. In Line 705 of 6. Please provide some perspectives on the role of this algorithm in the practical application of remote sensing data, which will be more conducive to future reference.

Finally, it is a nice research paper and congratulation on the authors’ work.

none.

Author Response

Dear Reviewer:

Thank you very much for your valuable feedback. We believe that your feedback has been very helpful in improving our paper. We have carefully read your feedback and revised the paper. The response to your feedback is in the file "Review response 1". Please review it and thank you again for your valuable feedback

Sincerely

Zongmin Zhao

Reviewer 2 Report

In this paper, the author propose a global structure and adaptive weight aware ICP algorithm, which not only solves the initial alignment problem of traditional ICP, but also has better computational complexity.

The principle of the algorithm is very detailed and clear with figures as explanations. And the authors conduct a substantial number of comparative analysis to confirm the efficacy and superiority of their algorithm. In the meantime, sufficient comparison experiments are conducted with appropriate experimental design. Also, The conclusions is consistent with the evidence and arguments presented.

But there is a problem with the subtitle of the paper, subtitle 5 is completely the same with subtitle 4, and the conclusion part should be the sixth part while you named it five. Besides, the expression of some sentences can be more rigorous, such as “The higher the curvature better dots are smaller ones” in line 242 and “The weight Wi(x) decreases with distance from point x on the surface” in line 254.

Author Response

Dear Reviewer:

Thank you very much for your valuable feedback. We believe that your feedback has been very helpful in improving our paper. We have carefully read your feedback and revised the paper. The response to your feedback is in the file "Review response 2". Please review it and thank you again for your valuable feedback

Sincerely

Zongmin Zhao

Reviewer 3 Report

In this paper the authors develop an algorithm to improve the iterative closest point (ICP) algorithm for point cloud or image registration. This algorithm uses the shape of the point cloud (global structure) and an algorithm to adapt weights attributed to registration points. The proposed method was compared to the initial ICP algorithm and to other 6 methods that improved the initial ICP algorithm. Two variables were proposed to measure the quality of the point cloud registration, the overlap area knockout value and the ground truth interior points. The proposed algorithm manages to better align point clouds when there are lot of noise, partial overlapping point clouds. The algorithm begins with a larger convergence domain compared to the ICP and is faster to reach the convergence point. It considers also the objects’ deformation and an adaptive weight iteration.

The paper presents a large literature review with in deep explanations of ICP algorithm and of the different algorithms that propose improvements. This explanation is useful for an external researcher to understand the main problem. A large number of equations are presented that sustain the algorithm description and help understanding. The results are presented and discussed in detail.

 Minor observations :

Line 19 : Give in the abstract the complete name for the “ICP” abbreviation.

Line 65 : Define the “RSICP”. Didn’t find the abbreviation on the cited paper [8].

Line 69 : Define the “FRICP” abbreviation. Didn’t find the abbreviation on the cited paper [9].

Lines 70-72 : This sentence is not clear. “it” refers to what algorithm?

Lines 76-77: “The classic ICP algorithm has a better convergence area and a faster convergence speed” compared to what? Compared to your proposed algorithm you just introduced in the sentence before? Why you propose a study if the classic algorithm is already better? The subject of this sentence should be your algorithm instead of the classic ICP?

Line 100 : “EPFL” Give the reference of this dataset.

Lines 149- 151 : This sentence is not clear, because it do not have a main verb. Please, reformulate.

Line 163 : Give the complete algorithm name for SIFT and RIFT algorithms.

Line 196: Give here the main reference for the classic ICP algorithm.

Lines 242-243: Sentence not clear.

Lines 245-2447: Sentence not clear, check the grammar.

Figure 7: Define the “injection point” in the text. Make a reference to this term in the text.

The paper is written in a good and understandable English.

However, some sentences must be revised.

Author Response

Dear Reviewer:

Thank you very much for your valuable feedback. We believe that your feedback has been very helpful in improving our paper. We have carefully read your feedback and revised the paper. The response to your feedback is in the file "Review response 3". Please review it and thank you again for your valuable feedback

Sincerely

Zongmin Zhao
